# Evaluating the Feasibility of the Liuxihe Model for Forecasting Inflow Flood to the Fengshuba Reservoir

**Yanjun Zhao, Yangbo Chen \*, Yanzheng Zhu and Shichao Xu**

School of Geography and Planning, Sun Yat-sen University, Guangzhou 510275, China
\* Correspondence: eescyb@mail.sysu.edu.cn; Tel.: +86-20-8411-4269

**Abstract:** Because of differences in the underlying surface, short flood confluence times, extreme precipitation, and other dynamic parameters, it is difficult to forecast an inflow flood to a basin reservoir, and traditional hydrological models do not achieve the forecast accuracy required for flood control operations. This study of the Fengshuba Reservoir in China evaluated the capacity of the Liuxihe model, which is based on a physically distributed hydrological model, to predict inflow floods in the Fengshuba Reservoir. The results show that the Liuxihe model has good applicability for flood forecasting in the basin. The use of different river classifications influenced the simulation results. The Liuxihe model can take into account the temporal and spatial inhomogeneity of precipitation and model parameters can be optimized using particle swarm optimization; this greatly improves the accuracy. The results show that the Liuxihe model can be used for real-time flood forecasting in the Fengshuba Reservoir watershed.

**Keywords:** Liuxihe model; flood forecast; river classification; forecast accuracy

## 1. Introduction

The basin flood forecasting models commonly used in China (and in other countries) can be divided into two types: lumped [1,2] and distributed models [3,4]. Representative lumped models include the Stanford [5], ARNO [6], and Xinanjiang models [7]; lumped models have the advantages of having fewer parameters, a faster computational speed, and their structures are easy to adapt to specific watershed characteristics. However, their reservoir inflow flood forecasting accuracy is not high [8,9]. The main reasons for this are as follows. First, these integrated models consider the watershed as a whole, i.e., the spatial characteristics within the watershed are not considered, and the models do not reflect the real watershed topography, soil cover, land type, or rainfall conditions [10]. Second, the flooding process is highly sensitive to rainfall, and integrated models cannot capture the spatial distribution of precipitation in the watershed [9]. In addition, integrated models require more historical flood data, and it is necessary to calibrate the parameters for each individual flood. For forecasting, the parameters closest to the historical flood data are selected but the error is relatively large. With the development of geographic information systems and remote sensing technologies, distributed hydrological models have been introduced, such as the SHE [11], VIC [12], Vflo [13], TOPMODEL [14–16], and Liuxihe models [17–21]. The TOPMODEL is a distributed hydrological model based on the topographic index and has been successfully applied for flood reservoir inflow forecasting in many headwater Asian catchments [22]. Distributed hydrological models consider changes in the underlying surface of the watershed, temporal and spatial changes in precipitation, and the impact of conservation projects on floods. Distributed hydrological models divide the watershed into grid units and treat the underlying surface as a collection of independent units that do not affect each other. Thus, the soil cover type, land use, and rainfall distribution differ among units, which is more representative of the actual situation. These models can represent flow generation and confluence in the entire

watershed, which greatly improves the accuracy of flood forecasting [23], but distributed hydrological models also face many problems such as too many input parameters and difficulty in optimization [24].

The Fengshuba Reservoir is located in Heyuan City, Guangdong Province (24°28′4.97″ N, 115°23′18.59″ E), and controls a drainage area of 5150 km². It is in a subtropical monsoon climate zone with a mild climate and abundant rainfall and sunshine. The annual average temperature is 19°C, and the average rainfall in the basin is 1560.9 mm. The Fengshuba Reservoir was designed according to a 1000-year flood, and the flood level for a 5000-year flood has been checked. The "checked flood level" is 172.7 m, the design flood level is 171.8 m, the normal storage level is 166 m, the total storage capacity is 1.932 billion m³, and the effective storage capacity is 1.25 billion m³. The reservoir was designed to meet the demand for flood control, water supply, irrigation, and power generation, and to facilitate shipping. It is an incomplete "annual regulation reservoir" that plays an important role in the regulation of water resources and flood control in nearby towns. Flood forecasting is key for flood control and dam safety, so it is necessary to establish a high-precision flood forecasting model [25].

The Fengshuba Reservoir controls a large watershed area, and the temporal and spatial distributions of rainfall within the area are uneven. It is difficult for lumped models to consider the impact of temporal and spatial changes in rainfall on flood formation, and forecasting accuracy cannot be guaranteed [10]. Moreover, reservoir construction is an intensely human activity that can influence the formation of inflow floods, such as by changing the local runoff mechanism (by increasing surface water storage and the confluence rate). The underlying surfaces in the Fengshuba Reservoir Basin show great spatial variation, as does the elevation; only distributed models can account for these effects [26–28]. Therefore, it is necessary to use a distributed hydrological model to accurately forecast inflow floods in the Fengshuba Reservoir.

The Liuxihe model, which is divided into fine grids, can fully consider the heterogeneity of the underlying surface and uneven spatial distribution of rainfall [29]. Because of the large number of parameters in the distributed hydrological model, the calculation requirements are very high. Thus, the Liuxihe model uses the particle swarm optimization (PSO) algorithm to optimize the parameters automatically [30]; there are 12 parameters that can be optimized in the Liuxihe model. For the practical application of the model, only one representative flood is needed for parameter optimization; the other flood data are used for verification. The Liuxihe model has successfully forecasted basin floods and provided mountain torrent disaster warnings [31,32], so it could theoretically be used for Fengshuba water inflow flood forecasting.

The purpose of this study is to evaluate the feasibility of the Liuxihe model for forecasting inflow floods of the Fengshuba Reservoir in Guangdong Province, China. The study adopts the Liuxihe model based on 90-m SRTM DEM data and the PSO algorithm is used to optimize the model parameters. The applicability of the Liuxihe model for flood forecasting of the Fengshuba Reservoir is evaluated and the effects of different river classifications on flood inflow simulation results are discussed. This study may be useful for scientists and practitioners who are involved in dam safety and flood risk reduction with the usage of flood inflow forecasting.

## 2. Data and Methods

### 2.1. Watershed Physical Characteristics

The physical characteristics data for the watershed required to build the model included digital elevation model (DEM), land use, and soil type data. The DEM data affect parameters such as the watershed area and slope and are thus needed for model construction. The data source used in this study was the Shuttle Radar Topography Mission (SRTM). SRTM data are available from a public database ("http://srtm.csi.cgiar.org/ (accessed on 23 September 2022)") at a spatial resolution of 90 m. The land use data, with a resolution of 1000 m, were from the United States Geological Survey (USGS) land-use type database

("http://landcover.usgs.gov/ (accessed on 20 September 2022)"). The soil-type data, which also had a resolution of 1000 m, were obtained from the Food and Agriculture Organization (FAO) soil-type database ("http://www.isric.org/ (accessed on 20 September 2022)"). The grid unit size used in the Liuxihe model was 90 m, and the land use and soil type data were resampled to 90 m. The results are shown in Figure 1. According to the land use and soil type data, there are eight types of land use in the Fengshuba Reservoir watershed, namely, evergreen coniferous forest, evergreen broad-leaved forest, shrub, rare forest, coastal wetland, slope grassland, lake, and cultivated land, which account for 39.75%, 12.63%, 36.32%, 2.42%, 0.09%, 3.08%, 0.56%, and 5.16% of the total land, respectively. Additionally, there are 20 soil types in the Fengshuba Reservoir Basin including iron-based low-activity strong-acid soil, simplified low-activity strong-acid soil, humus low-activity strong-acid soil, iron-aluminum prototype soil, simplified high-activity strong-acid soil, piled man-made soil, calcareous loose rock soil, humus rudimentary soil, and others. The DEM, land use, and soil type data of the Fengshuba watershed revealed obvious elevation changes within the watershed, spatial variability in land use, and heterogeneous physical characteristics between regions.

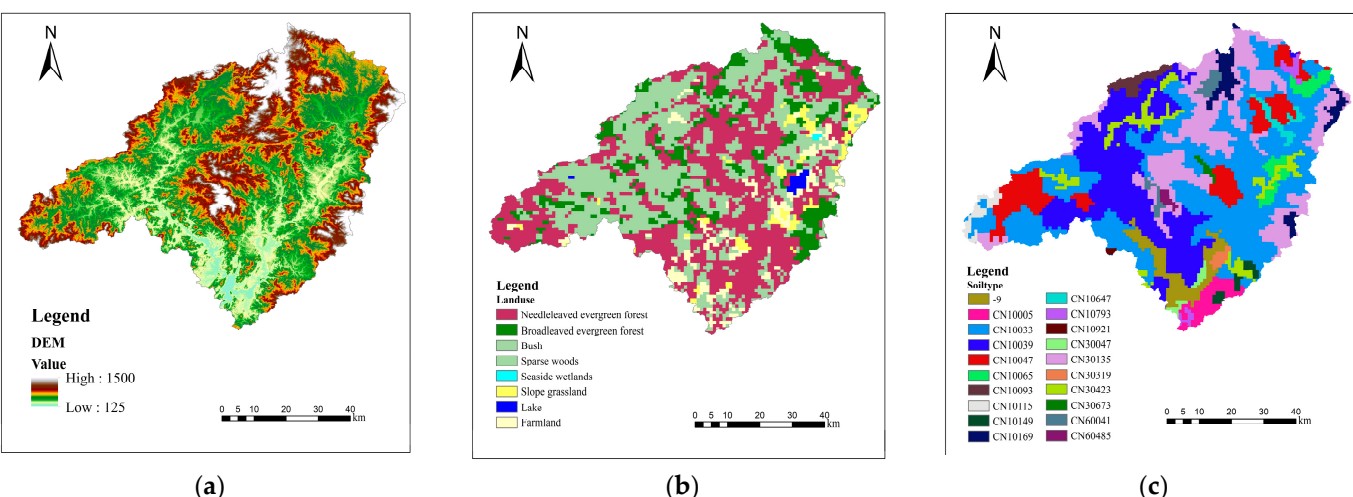

**Figure 1.** Globally available data of the Fengshuba Reservoir watershed. (**a**) DEM; (**b**) Land-use types; and (**c**) soil types.

*2.2. Hydrological Data*

There are 13 rainfall stations in the basin of the Fengshuba Reservoir, including the Sanheng and Longtang stations. The spatial positions of the rainfall stations and exit points of the Fengshuba Reservoir are shown in Figure 2. This study collected 18 datasets of flood processes from 2010 to 2020 in the Fengshuba Reservoir watershed, including rainfall-station rainfall data and inflow data with a time resolution of 1 h. Because the research area is the watershed of a reservoir, the inflow measurement is difficult to obtain due to the reservoir storage, and there is no flow station at the boundary of the reservoir return flow. Therefore, the flow data in this study were derived from a water level-capacity-discharge calculation, and relevant information about these flood events is shown in Table 1. The Liuxihe model is a distributed hydrological model with physical significance. One of its advantages over other models is that the model parameters are derived from terrain characteristics, and only one flood needs to be selected for parameter optimization [30]. The remaining floods (17 in this study) are used for simulation verification. The Thiessen polygon is used to interpolate the rainfall within the watershed. Thiessen polygon interpolation divides the watershed into different polygons according to the spatial position of each rainfall station, and the rainfall intensity within each polygon is determined by the rainfall intensities of the rain gauge stations within that polygon.

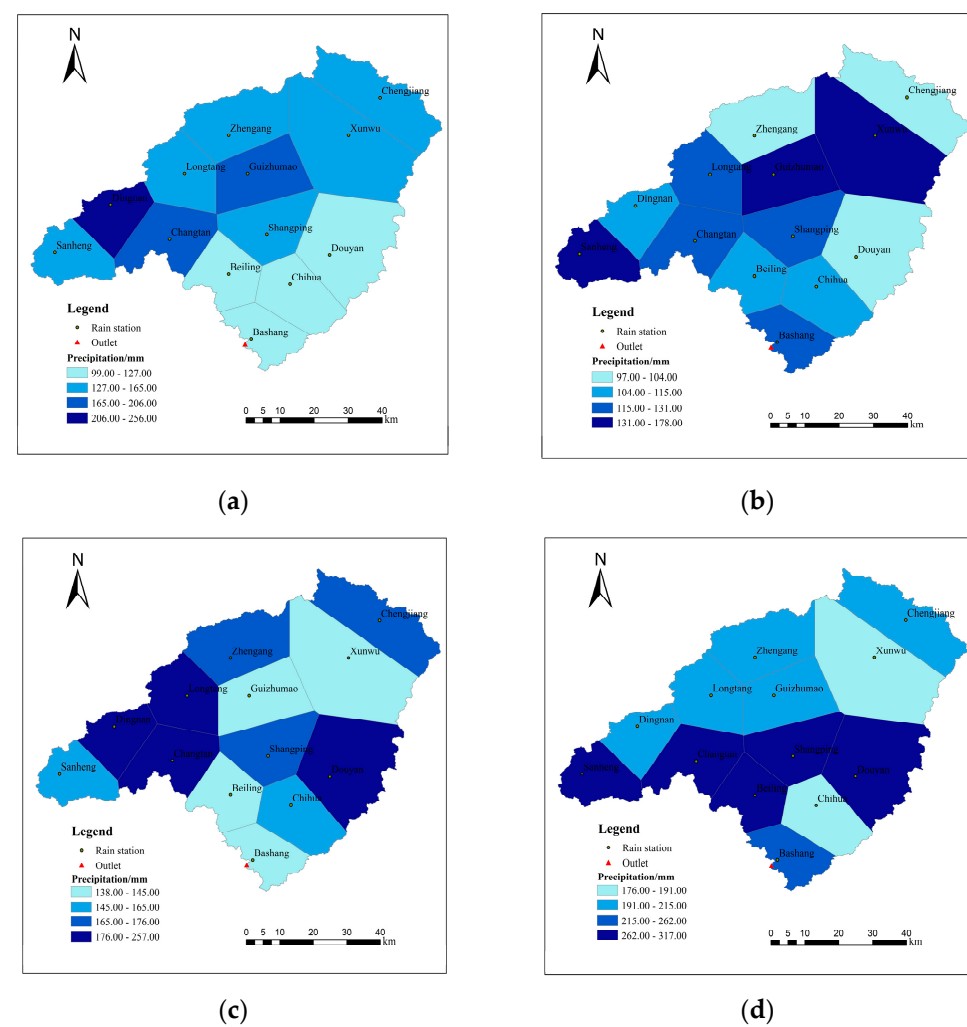

**Figure 2.** Rainfall distributions of randomly selected floods with flood numbers: (**a**) 2011051315; (**b**) 2014051901; (**c**) 2016042408; and (**d**) 2019060702.

**Table 1.** Information on the studied flood events.

| Flood Event No. | Start Time | End Time | Duration (h) | Total Rainfall (mm) | Peak Flow (m³/s) |
|---|---|---|---|---|---|
| 2010052211 | 22 May 2010 | 25 May 2010 | 62 | 775 | 1460 |
| 2011051315 | 13 May 2011 | 19 May 2011 | 153 | 2033 | 2170 |
| 2013051811 | 18 May 2013 | 24 May 2013 | 145 | 1503 | 1570 |
| 2014051901 | 19 May 2014 | 25 May 2014 | 167 | 1661 | 2850 |
| 2015052419 | 24 May 2015 | 28 May 2015 | 91 | 1037 | 1420 |
| 2016012613 | 26 Jan. 2016 | 4 Feb. 2016 | 204 | 2093.6 | 3030 |
| 2016031715 | 17 Mar. 2016 | 27 Mar. 2016 | 227 | 3094 | 3010 |
| 2016041006 | 10 April 2016 | 24 April 2016 | 337 | 2469 | 2990 |
| 2016042408 | 24 April 2016 | 8 May 2016 | 331 | 2322 | 3620 |
| 2016052003 | 20 May 2016 | 25 May 2016 | 118 | 1201 | 1490 |
| 2016101923 | 19 Oct. 2016 | 25 Oct. 2016 | 139 | 1334 | 1820 |
| 2016112501 | 25 Nov. 2016 | 30 Nov. 2016 | 126 | 782 | 1310 |
| 2017061201 | 12 June 2017 | 25 June 2017 | 326 | 2886 | 1480 |
| 2019041710 | 17 Apr. 2019 | 21 Apr. 2019 | 90 | 1052 | 1270 |
| 2019050417 | 4 May 2019 | 10 May 2019 | 135 | 1232 | 1060 |
| 2019060702 | 7 June 2019 | 19 June 2019 | 307 | 3149 | 4460 |
| 2019062012 | 20 June 2019 | 28 June 2019 | 183 | 1491.5 | 1830 |
| 2020060703 | 7 June 2020 | 12 June 2020 | 135 | 1326 | 1490 |

### 2.3. Temporal and Spatial Distribution of Rainfall

Precipitation is the main factor driving the formation of floods. The intensity of precipitation and temporal and spatial distribution of precipitation directly affect the magnitude and duration of a flood [33,34]. Temporal and spatial unevenness of precipitation is a common phenomenon, including in the basin; therefore, the spatial and temporal distribution of precipitation plays a decisive role in the predictive accuracy of models. The elevation, land use, and soil type are heterogeneous among different areas in the Fengshuba Reservoir Basin, and the precipitation in the watershed varies in time and space. In this study, the 18 field datasets for 2010–2020 collected during flooding were characterized by spatiotemporal inhomogeneity of precipitation. As shown in Figure 2, four floods were randomly selected, and analyzed in terms of the distribution of precipitation; the results revealed an uneven distribution of precipitation in the watershed.

## 3. Model Construction

### 3.1. Liuxihe Model Setup

The Liuxihe model divides the watershed into grid units for calculation purposes, so the size of the grid unit determines the calculation time of the model [35–37]. Generally, a smaller grid unit results in a more accurate representation of the surface type of the actual watershed and a longer model calculation time. Conversely, a larger grid unit results in a shorter model calculation time but lower accuracy. Therefore, it is necessary to find a compromise in terms of the resolution. The grid unit size used in the Liuxihe model for our study was 90 m. The Liuxihe model divides the watershed into grid units with independent physical characteristics and rainfall; each unit is divided into reservoir, channel, and slope units [38–40]. The river and reservoir units were divided according to the cumulative flow threshold and the threshold of the normal storage level; the remaining units in the basin were slope units.

The Liuxihe model has no clear preference for the classification of river units. It is generally recommended that small watersheds be divided into three levels of rivers and that large watersheds be divided into four levels. The watershed area of this study was medium-sized, so we evaluated the impact of third- and fourth-level rivers on the flood simulation of reservoir inflow and compared the flood process and simulation indicators of the two types of rivers. During the extraction of the river unit, rivers were classified according to the Strahler method [41]. According to different cumulative discharge thresholds, rivers were divided into grades 3 and 4. Finally, the number of third-level channel units was determined to be 5461, and the number of fourth-level channel units was 9457. According to the Google Maps remote sensing image, 25 channel nodes were set for the third-level channel, and the channel was divided into 41 virtual channel sections. Additionally, there were 88 nodes for the fourth-level channel, which divided the channel into 162 virtual channel sections. The channel section width, side slope, and bottom slope were estimated. The results of the third- and fourth-level segmentation node point construction are shown in Figure 3.

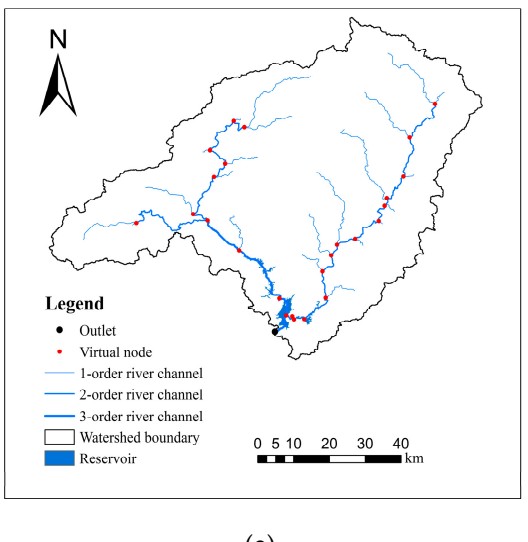

(**a**)

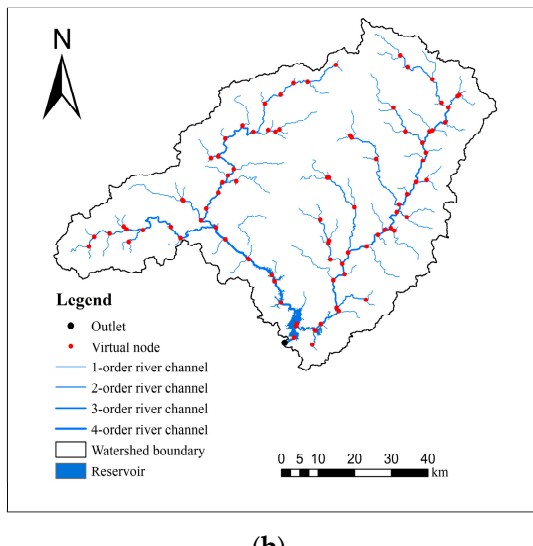

(**b**)

**Figure 3.** Liuxihe model structures with different channel classifications. (**a**) The model structure under a three-level channel and (**b**) the model structure under a four-level channel.

### 3.2. Derivation of Initial Parameters

The Liuxihe model divides the initial parameters into four categories: land use, soil, meteorological, and topographic parameters [17,42]. These initial parameters are determined by the corresponding physical characteristics of the grid cells.

1. Topographic parameters include the flow direction and slope and are obtained from the DEM data;
2. The main meteorological parameter is evaporation capacity. Based on experience, the evaporation capacity of all units was set at 5 mm/d [17];
3. The land use parameters include the slope roughness and evaporation coefficient, among which the evaporation coefficient is an insensitive parameter [17]. According to the parameterization experience of the Liuxihe model, the evaporation coefficient was uniformly set at 0.7 [17]. The slope roughness is a sensitive parameter, and the value in the recommended relevant literature was adopted [43,44], as shown in Table 2;

**Table 2.** Land use parameters.

| Land Use Type | Evaporation Coefficient | Slope Roughness Coefficient |
|---|---|---|
| Evergreen needle-leaf forest | 0.7 | 0.4 |
| Evergreen broadleaf forest | 0.7 | 0.6 |
| Bush | 0.7 | 0.4 |
| Sparse woods | 0.7 | 0.3 |
| Coastal wetland | 0.7 | 0.2 |
| Slope grassland | 0.7 | 0.1 |
| Lake | 0.7 | 0.2 |
| Farmland | 0.7 | 0.15 |

4. Soil parameters include the saturated water content, saturated hydraulic conductivity, field water holding rate, wilting water content, soil thickness, and soil properties. The value of soil properties was set uniformly to 2.5 [17], and the other parameters were calculated using the soil hydraulic characteristic calculator proposed by Arya et al. [45]. The results are shown in Table 3.

**Table 3.** Soil-type parameters.

| Soil Type | Thickness of Soil Layer (mm) | Saturated Water Content | Field Moisture Retention | Saturated Hydraulic Conductivity (mm·h$^{-1}$) | Soil Characteristic Coefficient | Wilting Moisture Content |
|---|---|---|---|---|---|---|
| CN-9 | 0.0001 | 0.0001 | 0.0001 | 0.0001 | 2.5 | 0.0001 |
| CN10005 | 1000 | 0.502 | 0.355 | 9.82 | 2.5 | 0.136 |
| CN10033 | 1000 | 0.451 | 0.3 | 8.64 | 2.5 | 0.176 |
| CN10039 | 600 | 0.515 | 0.422 | 1.95 | 2.5 | 0.296 |
| CN10047 | 1000 | 0.455 | 0.319 | 6.34 | 2.5 | 0.192 |
| CN10065 | 1000 | 0.491 | 0.433 | 0.47 | 2.5 | 0.315 |
| CN10093 | 1000 | 0.454 | 0.144 | 74.49 | 2.5 | 0.063 |
| CN10115 | 700 | 0.500 | 0.377 | 4.89 | 2.5 | 0.221 |
| CN10149 | 1000 | 0.481 | 0.390 | 1.86 | 2.5 | 0.262 |
| CN10169 | 1000 | 0.458 | 0.252 | 23.82 | 2.5 | 0.110 |
| CN10647 | 1000 | 0.454 | 0.337 | 3.99 | 2.5 | 0.214 |
| CN10793 | 1110 | 0.436 | 0.249 | 15.76 | 2.5 | 0.149 |
| CN10921 | 1000 | 0.495 | 0.391 | 2.78 | 2.5 | 0.255 |
| CN30047 | 1500 | 0.461 | 0.265 | 20.78 | 2.5 | 0.115 |
| CN30135 | 1000 | 0.435 | 0.207 | 28.33 | 2.5 | 0.121 |
| CN30319 | 800 | 0.453 | 0.239 | 26.07 | 2.5 | 0.109 |
| CN30423 | 670 | 0.446 | 0.240 | 21.87 | 2.5 | 0.126 |
| CN30673 | 1000 | 0.443 | 0.201 | 29.31 | 2.5 | 0.121 |
| CN60041 | 870 | 0.438 | 0.260 | 13.86 | 2.5 | 0.154 |
| CN60485 | 250 | 0.470 | 0.323 | 8.38 | 2.5 | 0.175 |

The Liuxihe model focuses on studying time periods with large changes in river flow; the impact of base flow on floods is relatively small during flood periods, but the Liuxihe model takes base flow into account and calculates it through groundwater [17]. In addition, initial conditions for the catchment area will be set before simulation, including initial soil moisture content. These initial conditions are determined based on the recommended values [17].

*3.3. Parameter Optimization Method*

In this study, the PSO algorithm [30] was used to optimize the initial parameters of the model. The PSO algorithm is a global optimization algorithm that simulates the foraging behavior of birds to find the optimal destination through collective information sharing; the algorithm has the advantages of a fast convergence speed and high efficiency.

Every individual particle represents a possible solution to the model parameters, and selecting the optimal number of particles is a critical PSO parameter that can significantly affect the performance of the PSO algorithm [17]. These particles advance simultaneously across the search space in accordance with specific rules, which can be defined using the equations provided below.

$$V_{i,\ k} = w \times V_{i,k-1} + C1 \times rand \times \left( X_{i,\ pBest} - X_{i,k-1} \right) + C2 \times rand \times \left( X_{gBest} - X_{i,k-1} \right) \quad (1)$$

$$X_{i,k} = X_{i,k-1} + X_{i,k} \quad (2)$$

where $V_{i,\ k}$ is the moving speed of the *i*th particle at the *k*th step, $X_{i,k}$ is the position of the *i*th particle at the *k*th step, $X_{i,\ pBest}$ is the best position of the *i*th particle at the *k*th step (current), $X_{gBest}$ is the best position of all particles at the *k*th step, $w$ is the inertia acceleration speed, $C1$ and $C2$ are learning factors, and *rand* is a random number between 0 and 1.

$w$, $C1$, and $C2$ are important PSO parameters that will impact the PSO's performance. The global search capability is influenced by the inertia weight $w$, which is dynamically determined using a linearly decreasing inertia weight (LDIW) approach based on the following equation:

$$w = w_{max} - \frac{i \left( w_{max} - w_{min} \right)}{MaxN} \quad (3)$$

where $i$ is the current evolution number, $MaxN$ is the maximum evolution number, $w_{max}$ takes the value of 0.9, and $\omega_{min}$ takes the value of 0.1.

The arccosine function strategy is employed to determine the values of $C1$ and $C2$; the equations are listed below.

$$C1 = C1_{min} + (C1_{max} - C1_{min}) \left( 1 - \frac{arccos \left( \frac{-2 \times i}{MaxN} + 1 \right)}{\pi} \right) \tag{4}$$

$$C2 = C2_{min} + (C2_{max} - C2_{min}) \left( 1 - \frac{arccos \left( \frac{-2 \times i}{MaxN} + 1 \right)}{\pi} \right) \tag{5}$$

where $C1_{max}$ and $C1_{min}$ are the maximum and minimum values of $C1$, and the values of 2.75 and 1.25 are recommended. $C2_{max}$ and $C2_{min}$ are the maximum and minimum values of $C2$, and the values of 2.5 and 0.5 are recommended. $i$ is the current evolution number and $MaxN$ is the maximum evolution number.

### 3.4. Model Valiadtion

The performance of the model simulation is evaluated with six indexes, including the Nash–Sutcliffe coefficient $NSE$, correlation coefficient $R$, peak flow relative error $E$, peak discharge delay $\Delta H$, process relative error $PRE$, and water balance coefficient $WBC$, which are calculated below.

The correlation coefficient is employed to assess the extent of scattering between the simulated and observed flow.

$$R = \frac{N \sum_{i=1}^{N} Q_{obs}^i Q_{sim}^i - \sum_{i=1}^{N} Q_{obs}^i \sum_{i=1}^{N} Q_{sim}^i}{\sqrt{\left[ N \sum (Q_{obs}^i)^2 - \left( \sum_{i=1}^{N} Q_{obs}^i \right)^2 \right] \left[ N \sum (Q_{obs}^i)^2 - \left( \sum_{i=1}^{N} Q_{sim}^i \right)^2 \right]}} \tag{6}$$

where $Q_{sim}^i$ and $Q_{obs}^i$ are the simulated and observed flow at the time $i$, respectively, and $N$ is the total time steps for a simulated flood event.

The Nash–Sutcliffe coefficient is used to evaluate the accuracy of the model simulation results and reflect the overall fit of the model.

$$NSE = 1 - \frac{MSE^2}{F_0^2} \tag{7}$$

$$MSE = \sqrt{\frac{1}{N} \sum_{i=1}^{N} \left( Q_{sim}^i - Q_{obs}^i \right)^2} \tag{8}$$

$$F_0 = \sqrt{\frac{1}{N} \sum_{i=1}^{N} \left( Q_{obs}^i - \overline{Q_{obs}^i} \right)^2} \tag{9}$$

Process relative error is used to evaluate the degree to which the simulated value deviates from the actual observed value.

$$PRE = \frac{1}{N} \sum_{i=1}^{N} \frac{\left| Q_{obs}^i - Q_{sim}^i \right|}{Q_{obs}^i} \tag{10}$$

The water balance coefficient is used to evaluate the ratio of the total water error simulated by the model to the total runoff.

$$WBC = \frac{\sum_{i=1}^{N} Q_{sim}^i}{\sum_{i=1}^{N} Q_{obs}^i} \tag{11}$$

Peak flow relative error is used to evaluate the coincidence between the observed and simulated flood peak discharge.

$$E = \frac{QP_{sim} - QP_{obs}}{QP_{obs}} \tag{12}$$

where $QP_{sim}$ is the simulated peak flow and $QP_{obs}$ is the observed peak flow.

Peak discharge delay is used to evaluate the time interval between the occurrence of the actual flood peak and the occurrence of the simulated flood peak.

$$\Delta H = HP_{sim} - HP_{obs} \tag{13}$$

where $HP_{sim}$ is the time (hour) that the simulated peak flow occurred, and $HP_{obs}$ is the time (hour) that the peak flow is observed.

## 4. Results

### 4.1. Parameter Optimization of the Liuxihe Model

Based on the PSO algorithm, the Liuxihe model only needs to optimize parameters for one flood. The remaining floods are used for simulation verification. In this study, flood event number 2016042408 was used to optimize the parameters of the Liuxihe model, and the remaining 17 floods were used for simulation verification. In the algorithm, the number of particles was set to 20, the number of iterations was 50, and the total number of calculations was 1000. Finally, the optimization run time of the PSO algorithm was 20 h. Figure 4 shows the evolution of the objective function and parameter values during the optimization process of the four-level channel model parameters, and the simulated hydrological diagram before and after parameter optimization. The results showed that the model objective function value tended to be stable and that the model parameters converged to the optimal state after 50 parameter iterations. They also showed that the parameters of the Liuxihe model had a good convergence speed and that the simulation results were very close to the measured ones. The flooding process after parameter optimization fit the measured flood process well. The Nash–Sutcliffe coefficient was 0.715, the correlation coefficient was 0.876, the flood peak error was 0.013, and the flood peak time error was −1.

### 4.2. Model Performance Evaluation

One of the advantages of the Liuxihe model is that only one flood is used for parameter optimization, and the remaining flood data are used for simulation verification. The average indicators of the flood verification simulations are shown in Table 4.

**Table 4.** Statistical indicators of the flood simulation results after parameter optimization.

| Parameter | Average Nash–Sutcliffe Coefficient | Average Correlation Coefficient | Average Process Relative Error | Average Peak Error | Average Water Balance Coefficient | Average Peak Time Error |
|---|---|---|---|---|---|---|
| Optimized | 0.58 | 0.85 | 0.65 | 0.03 | 0.98 | 2.8 |

In the 17 simulations, the flood peak error was <0.2, the average flood peak error was 0.03, and the average flood peak time error was 2.8. According to the regulation of hydrological forecasting, the allowable error of a rainfall-runoff forecast is 20% of the actual measured flood peak discharge. That is, the flood peak error is within 0.2, and the time of flood peak occurrence is within 3 h of the actual occurrence. According to the standards, the Liuxihe model forecasting scheme constructed in this study is suitable for the real-time forecasting of inflow floods in the Fengshuba Reservoir Basin.

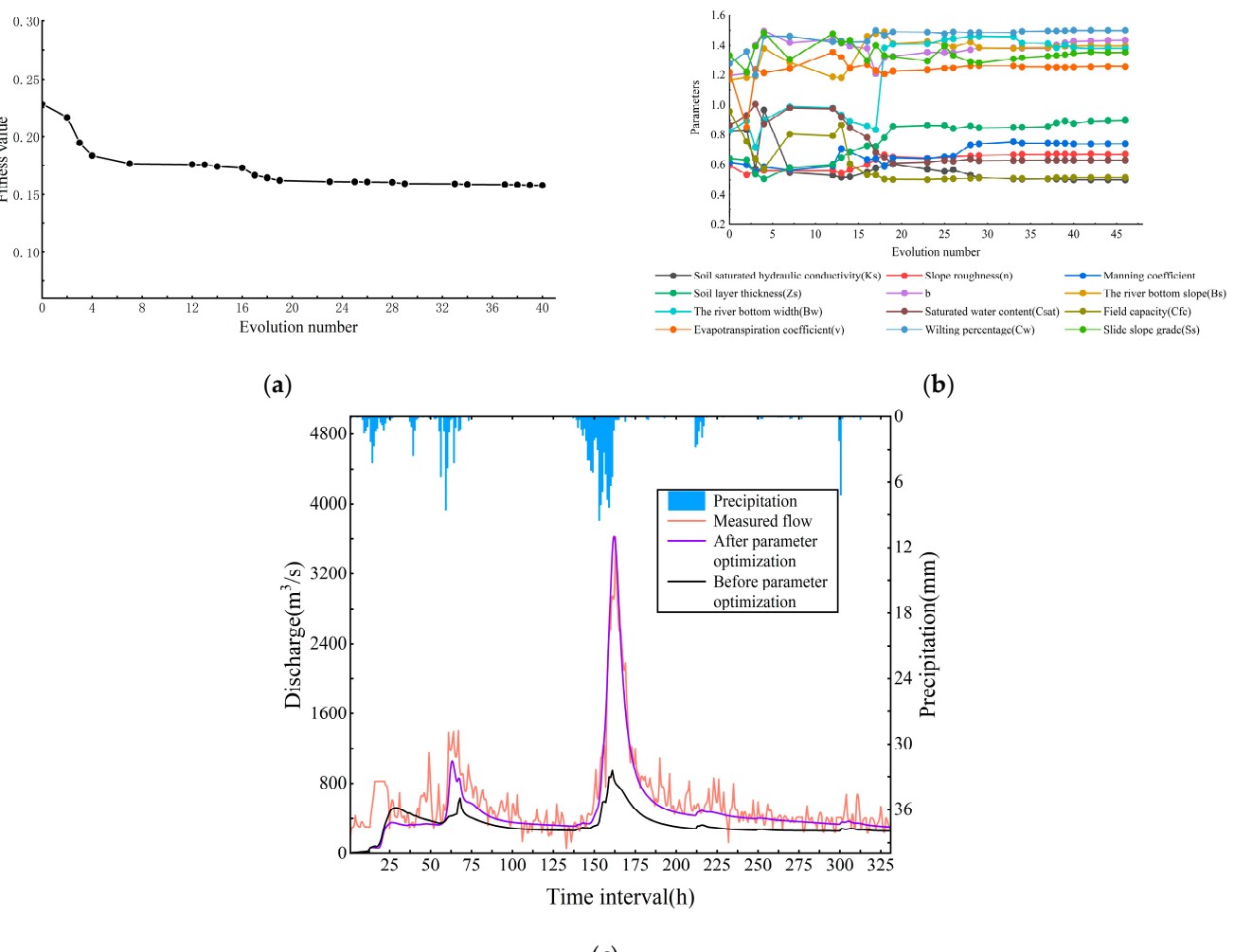

**Figure 4.** Results of the parameter optimization of the Liuxihe model with the PSO algorithm. (**a**) The changing curve of the objective function; (**b**) the parameter evolution process; and (**c**) the simulated hydrographs before and after parameter optimization.

This study ranks 18 selected floods according to their peak discharge, considering the top 40% as large flood events, and takes into account the characteristics of the flood event (e.g., double peaks), because large flood events are more catastrophic and double peaks are more complex than a single peak, so that 6 representative flood simulation results were selected and are shown in Figure 5.

The time distribution of precipitation directly affects the size and duration of floods. Additionally, the Fengshuba Reservoir Basin has the characteristic of uneven rainfall in both space and time. In the 18 flood datasets collected in this research, the events were mainly caused by concentrated or extreme precipitation. One of the advantages of the Liuxihe model is that the PSO algorithm is used to select a flood event for parameter optimization, and the remaining flood events are simulated and verified. As shown in Figure 5, flood numbers 2016041006 and 2019060702 have uneven precipitation, but the model exhibited good simulation performance. The error and time error of the flood peak are both small, which shows that the Liuxihe model can simulate the effect of the actual flood in the case of uneven precipitation. As a distributed hydrological model, the Liuxihe model divides the watershed into independent unit grids. Each grid unit has independent physical characteristics and rainfall distribution. The Liuxihe model can consider the heterogeneity between grid units and thus has better precision.

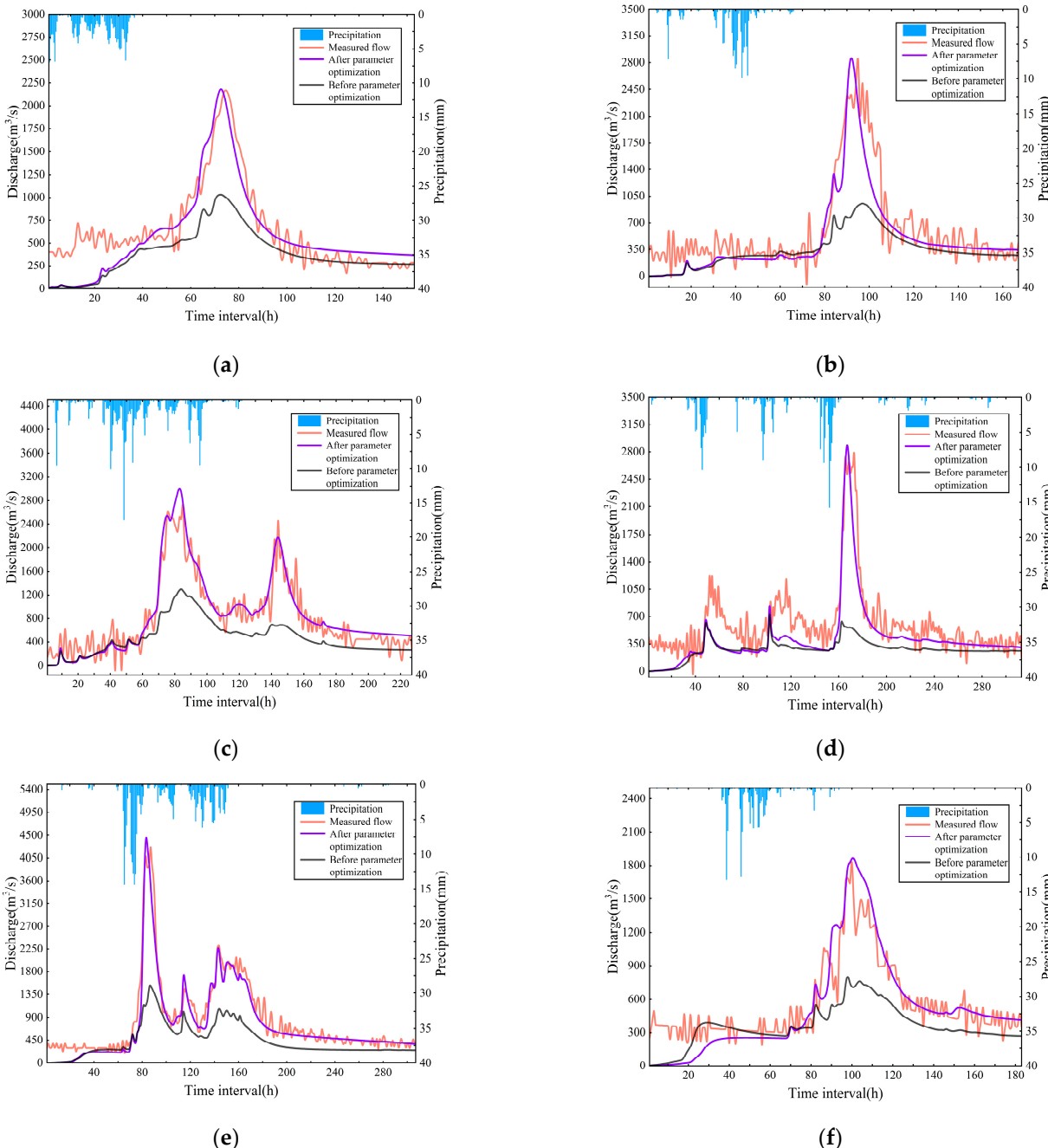

**Figure 5.** Representative floods with flood numbers: (**a**) 2011051315; (**b**) 2014051901; (**c**) 2016031715; (**d**) 2016041006; (**e**) 2019060702; and (**f**) 2019062017.

In addition, it can be observed from Figure 5 that the initial simulated value is zero, this is because the model sets the time before the flood as a pre-heating buffer to improve the performance of the model.

### 4.3. Influence of Parameter Optimization

To analyze the performance of the model after parameter optimization by the PSO algorithm, 17 floods were simulated using the initial and optimized model parameters. The average indicators of the flood simulation results are shown in Table 5, and six representative flood cases before and after parameter optimization are presented in Figure 5.

**Table 5.** Statistical indicators of the flood simulation results before and after parameter optimization.

| Parameter | Average Nash–Sutcliffe Coefficient | Average Correlation Coefficient | Average Process Relative Error | Average Peak Error | Average Water Balance Coefficient | Average Peak Time Error |
|---|---|---|---|---|---|---|
| Optimized | 0.58 | 0.85 | 0.65 | 0.03 | 0.98 | 2.8 |
| Initial | 0.27 | 0.76 | 0.64 | 0.49 | 0.69 | −3.33 |

The results showed that the simulation accuracy of the Liuxihe model was significantly improved after parameter optimization. The Liuxihe model is a distributed hydrological model, and its parameters are determined by the physical characteristics of the surface in the watershed. Therefore, the model includes uncertainties. However, there is a set of parameter values that in theory results in the highest model simulation accuracy; this value set represents the optimal parameters. When there is no parameter optimization, the Liuxihe model has uncertainties in flood simulation and forecasting. However, when the parameters of the Liuxihe model were optimized through the PSO algorithm, the flood simulation results using the optimized model showed that the average Nash–Sutcliffe coefficient increased by 31%, the average peak flow error decreased by 46%, the average correlation coefficient increased by 9%, and the average peak flow occurrence time error decreased. After parameter optimization, the flood simulation effect of the Liuxihe model was significantly improved in comparison with the simulation effect of the initial parameters, which further demonstrates that the distributed hydrological model can improve model performance through parameter optimization.

After optimizing the parameters with the PSO algorithm, the Liuxihe model was used to simulate 17 floods. The results showed that the rate of a flood peak error < 20% was 100% and the rate of a peak flow occurrence time error < 3 h was 76.5%.

### 4.4. Influence of River Classification on Simulation Results

River units affect the process of surface runoff confluence. Flood number 2016042408 was selected as a case study for optimizing the parameters of the third- and fourth-level rivers, and the remaining 17 floods were used for simulation verification. The input parameters were the same, except for the channel classification. The optimized parameters with the large change from the three-level channel to the four-level channel model are shown in Table 6. The remaining parameters have little difference, including soil saturation, hydraulic conductivity, soil layer thickness, soil characteristic coefficient, field capacity, wilting percentage, side slope grade, potential evaporation rate, and subsurface runoff coefficient.

**Table 6.** Parameter results of the three- and four-level channel optimizations.

| Parameters | Saturated Water Content (Csat) | Slope Roughness (n) | Manning Coefficient | Evaporation Coefficient (v) | River Bottom Slope (Bs) | River Bottom Width (Bw) |
|---|---|---|---|---|---|---|
| 2016042408 (4-level) | 0.629 | 0.67 | 0.738 | 1.255 | 1.397 | 1.381 |
| 2016042408 (3-level) | 1.343 | 1.232 | 1.496 | 0.542 | 0.5 | 0.771 |

To compare the effects of the two river classes on the simulation of reservoir inflow, the optimized parameters of the third- and fourth-class rivers were used to simulate and verify the remaining 17 flood cases. The average statistical indicators of the simulation results are shown in Table 7.

**Table 7.** Comparison of statistical indicators for the flood simulation in different river classes.

| Flood Event Number | NSE | R | PRE | E | WBC | ΔH (h) |
|---|---|---|---|---|---|---|
| 2010052211 | 0.593 | 0.837 | 0.399 | 0.016 | 0.922 | 0 |
| | 0.572 | 0.814 | 0.393 | 0.061 | 0.864 | −1 |
| 2011051315 | 0.705 | 0.864 | 0.569 | 0.008 | 0.91 | −1 |
| | 0.62 | 0.808 | 0.655 | 0.021 | 0.886 | −4 |
| 2013051811 | 0.482 | 0.776 | 0.338 | 0.061 | 0.892 | 20 |
| | 0.497 | 0.778 | 0.327 | 0.072 | 0.898 | 12 |
| 2014051901 | 0.784 | 0.908 | 0.432 | 0.016 | 0.803 | −3 |
| | 0.84 | 0.924 | 0.432 | 0.003 | 0.932 | −3 |
| 2015052419 | 0.334 | 0.816 | 0.487 | 0.012 | 0.691 | −2 |
| | 0.484 | 0.805 | 0.425 | 0.002 | 0.809 | 2 |
| 2016012613 | 0.298 | 0.765 | 1.542 | 0.022 | 1.391 | −1 |
| | 0.484 | 0.762 | 0.968 | 0.065 | 1.248 | −3 |
| 2016031715 | 0.768 | 0.889 | 0.968 | 0.002 | 1.043 | −2 |
| | 0.683 | 0.837 | 1.204 | 0.132 | 1.081 | −6 |
| 2016041006 | 0.569 | 0.847 | 0.644 | 0.025 | 0.697 | −6 |
| | 0.522 | 0.776 | 0.777 | 0.056 | 0.845 | −9 |
| 2016042408 | 0.715 | 0.876 | 0.667 | 0.013 | 0.844 | −1 |
| | 0.572 | 0.82 | 0.742 | 0.016 | 0.759 | −2 |
| 2016052003 | 0.593 | 0.88 | 0.476 | 0.024 | 0.876 | 2 |
| | 0.406 | 0.771 | 0.524 | 0.023 | 0.849 | −3 |
| 2016101923 | 0.465 | 0.789 | 0.634 | 0.068 | 1.154 | −2 |
| | 0.423 | 0.751 | 0.731 | 0.057 | 1.225 | −4 |
| 2016112501 | 0.11 | 0.714 | 0.906 | 0.019 | 1.25 | 3 |
| | 0 | 0.699 | 1.037 | 0.014 | 1.377 | 3 |
| 2017061201 | 0.5 | 0.74 | 2.002 | 0.091 | 0.961 | 44 |
| | 0.259 | 0.574 | 2.411 | 0.077 | 0.943 | 42 |
| 2019041710 | 0.763 | 0.92 | 0.464 | 0.047 | 1.026 | 5 |
| | 0.632 | 0.812 | 0.573 | 0.01 | 0.991 | 2 |
| 2019050417 | 0.487 | 0.859 | 0.312 | 0.06 | 1.305 | 3 |
| | 0.467 | 0.799 | 0.38 | 0.003 | 1.116 | 0 |
| 2019060702 | 0.913 | 0.964 | 0.254 | 0.01 | 0.887 | −4 |
| | 0.83 | 0.92 | 0.343 | 0.03 | 1.05 | −1 |
| 2019062012 | 0.679 | 0.918 | 0.32 | 0.02 | 0.944 | 0 |
| | 0.629 | 0.869 | 0.317 | 0.022 | 0.783 | −3 |
| 2020060703 | 0.659 | 0.87 | 0.333 | 0.023 | 0.987 | −3 |
| | 0.335 | 0.689 | 0.37 | 0.03 | 0.798 | −4 |
| Average indicator | 0.579 | 0.846 | 0.652 | 0.029 | 0.976 | 2.8 |
| | 0.514 | 0.789 | 0.7 | 0.039 | 0.97 | 1 |

The results show that the average values of the statistical indicators of the fourth-grade rivers are better than those of the third-grade rivers. The average values of the Nash–Sutcliffe coefficient of the fourth-grade rivers and third-grade rivers are 0.579 and 0.514, respectively, reflecting an improvement of 6.5%. The average correlation coefficients are 0.846 and 0.789, respectively (an increase of 5.7%). Additionally, the process relative error decreased by 4.8%, and the average peak error decreased by 1%. The PSO algorithm improves the simulation accuracy of the Liuxihe model as well as the convenience of the Liuxihe model in practical applications where only a single typical flood is needed for parameter optimization. Four flood simulation hydrological maps are shown in Figure 6.

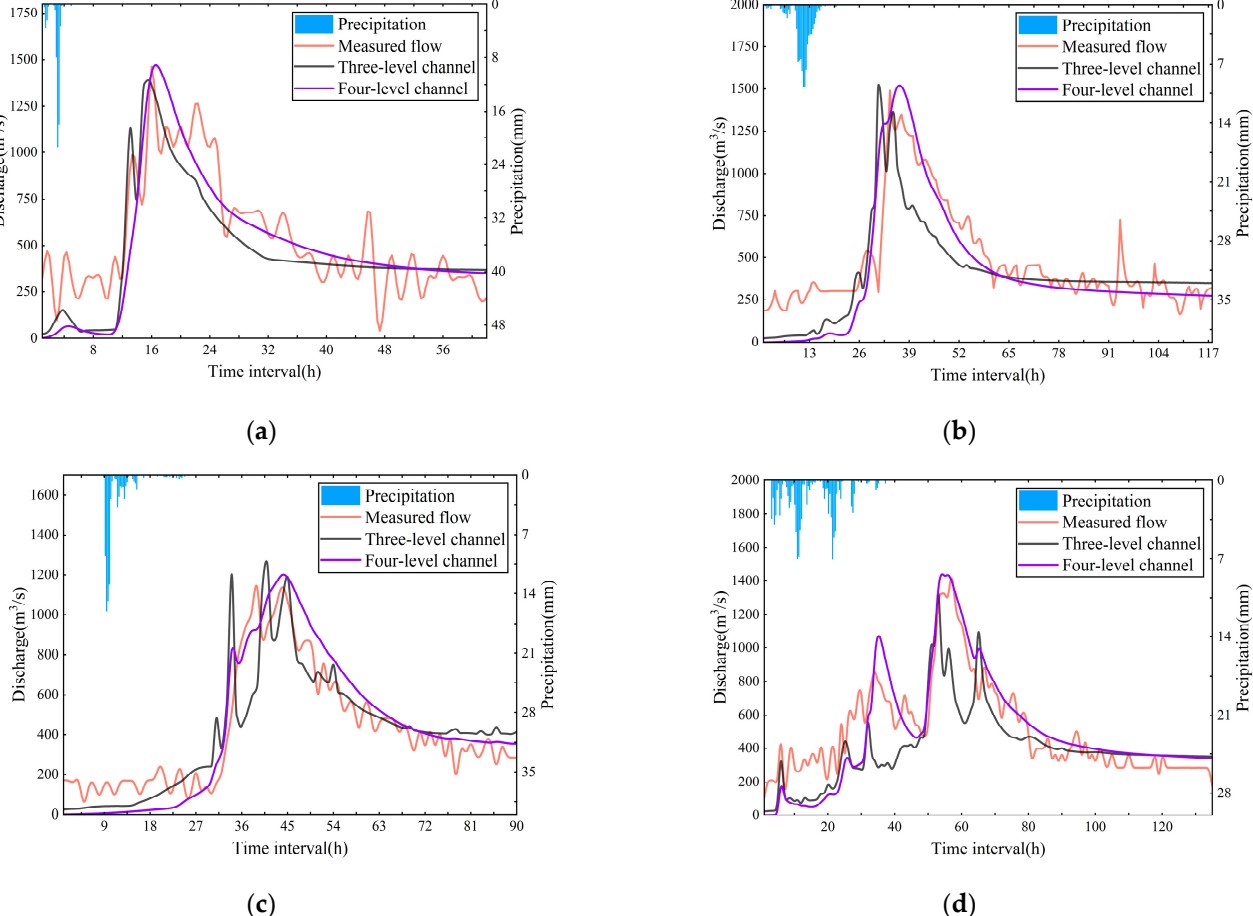

**Figure 6.** Simulation processes of representative floods with flood number: (**a**) 2010052211; (**b**) 2016052003; (**c**) 2019041710; and (**d**) 2020060703.

The results show that the simulation effect of the fourth-order river is better than the effect of the third-order river and that the peak error of the fourth-order river is lower than the peak error of the third-order river.

According to the statistical indicators of the simulation results and flood process line diagram, in comparison with the third-level channel, the peak value of the fourth-level channel simulation result is closer to the peak value of the measured flood, the corresponding statistical indicators are optimized, and the flood simulation accuracy is improved; therefore, it is more appropriate to use four-level channel modeling for Fengshuba Reservoir inflow flood forecasting. The average accuracy of this model is higher than that of the third-level channel, and most of the flood simulation curves are more consistent with the measured results.

## 5. Discussion

In this study, we utilized the Liuxihe model to simulate 18 flood events occurring in the Fengshuba Reservoir watershed. Our findings suggest that the Liuxihe model performs well in predicting the inflow of the reservoir. However, certain issues still require improvement, including:

1. Rainfall is a key factor in the formation of floods, and the quality of rainfall interpolation methods can affect the amount of precipitation on the surface of the basin. The Liuxihe model uses the most widely used and common rainfall data interpolation technique (Thiessen polygons). Therefore, in order to approach the true precipitation situation on the surface of the basin, improvements are needed in the rainfall interpolation method.

2.  We performed a simulation analysis without considering the operation of the upstream watershed of the Fengshuba Reservoir. The reservoir flow simulation modeled the natural runoff and confluence within the basin, which eventually reaches Fengshuba Reservoir without factoring in the reservoir's impact, but the reality is that it will be affected by the operation of the reservoir.

## 6. Conclusions

This study of the Fengshuba Reservoir watershed used the Liuxihe model to simulate a reservoir inflow flood and evaluated the utility of the Liuxihe model to predict reservoir inflow floods in terms of model simulation accuracy, parameter selection, and river classification. We also considered the impact of the temporal and spatial heterogeneity of rainfall on the model. The study produced several interesting results:

1.  The Liuxihe model showed good simulation accuracy for reservoir inflow floods. The average error of the flood peak was <0.02, and the average error of the flood peak time was <3 h. The Liuxihe model was found to be suitable for flood forecasting in the Fengshuba Reservoir Basin. The statistical index values of some simulations were low because the measured flood flow was highly irregular; fluctuations and outliers existed.
2.  The initial parameters of the model were uncertain, but the simulation performance of the Liuxihe model was improved significantly through parameter optimization. After parameter optimization, the average Nash–Sutcliffe coefficient was 31% higher, the average peak flow error was 46% lower, the average correlation coefficient was 9% higher, and the average peak flow time error was reduced.
3.  The influence of different river classifications on the model was examined. Compared with that of a third-class river, the simulation performance of the Liuxihe model constructed using a fourth-class river was better: there was an increase in the average value of the Nash–Sutcliffe coefficient by 6.5% and the average value of the correlation coefficient by 5.7% as well as a decrease in the process relative error by 4.8% and the average peak error by 1%.
4.  The distribution of precipitation in the watershed is uneven in time and space. However, the Liuxihe model can still simulate the uneven distribution of precipitation with high accuracy.

**Author Contributions:** Y.C. was responsible for proposing the original idea and providing technical guidance; Y.Z. (Yanjun Zhao) was responsible for the data compilation, processing, computation, and writing; Y.Z. (Yanzheng Zhu) and S.X. were responsible for the data sorting. All authors have read and agreed to the published version of the manuscript.

**Funding:** This study was supported by the National Natural Science Foundation of China (NSFC) (no. 51961125206) and the Science and Technology Program of Guangdong Province (no. 2020B1515120079).

**Data Availability Statement:** Data sharing is not applicable.

**Conflicts of Interest:** The authors declare no conflict of interest.

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
