# Peer review of "Evaluating the Feasibility of the Liuxihe Model for Forecasting Inflow Flood to the Fengshuba Reservoir"

_water, doi:10.3390/w15061048_

Round 1

Reviewer 1 Report

Thank you for the opportunity to review the manuscript by Zhao et al. titled “Evaluating the Feasibility of the Liuxihe Model for Forecasting Inflow Flood to the Fengshuba Reservoir”. The authors simulate 18 flood inflows to the dam reservoir from past precipitation events in the upstream catchment of the Fengshuba Reservoir. The study may be useful for scientists and practitioners who are involved in dam safety and flood risk reduction with the usage of flood inflow forecasting. However, the manuscript is poorly structured and lack important equations of  Liuxihe Model governing equations and equations of statistical indicators utilized by the authors. Therefore, the manuscript requires revisions before it can be accepted to water, see my concerns and comments below.

Lines 22-34: This paragraph is out of place and should be moved to study area description (or in Line 63). The authors should also add the missing information about the Fengshuba Dam: 1) controlled and uncontrolled flood discharge at  and 2) the 1000-year flood discharge that was used for the Fengshuba Dam design.

Line 48: The authors need to add three references that describe one of the mostly used distributed hydrological models based on the topographic index and applied for flood reservoir inflow forecasting in many headwater Asian catchments:

Beven, K.J., Kirkby, M.J., Freer, J.E., Lamb, R. A history of TOPMODEL. Hydrol. Earth Syst. Sci., 25, 527–549, 2021, https://doi.org/10.5194/hess-25-527-2021

Navarathinam, K., Gusyev, M., Magome, J., Hasegawa, A., Takeuchi, K. Agricultural flood and drought risk reduction by a proposed multi-purpose dam: A case study of the Malwathoya River Basin, Sri Lanka. In Weber, T., McPhee, M.J. and Anderssen, R.S. (eds) MODSIM2015, 21st International Congress on Modelling and Simulation. Modelling and Simulation Society of Australia and New Zealand, 2015, 1600-1606, ISBN: 978-0-9872143-5-5. 

Okazumi, T., Shrestha, B., Miyamoto, M., Gusyev, M. Uncertainty Estimation during the Process of Flood Risk Assessment in Developing Countries – Case study in the Pampanga River Basin–. J. of Disaster Research 2014, 9, 69-77.

Liu, L., Ao T., Zhou, L., Takeuchi, K., Gusyev, M., Zhang, X., Wang, W., Renhi, Y. Comprehensive evaluation of parameter importance and optimization based on the integrated sensitivity analysis system: A case study of the BTOP model in the upper Min River Basin, China. J. of Hydrology, 2022, 610, 127819, https://doi.org/10.1016/j.jhydrol.2022.127819

Line 49: The key reference of Liuxihe model should be added

Hutting, R.J.M. (2007) Hydrological modelling of the Liuxihe River basin to contribute to the development of flood management. Master Thesis, University of Twente. https://purl.utwente.nl/essays/687

Lines 66-75: This text is out place and it should be made a new paragraph. The authors should also indicate the number of parameters utilized in the model calibration.

Lines 76-81: This paragraph lacks a description of the manuscript and should be completely re-written to include the authors’ investigations in the sections of the manuscript.

Lines 104-111: This statement should be moved to model setup and the authors make a new sub-section in Section 3 that should provide the governing equations with parameters that are used for the Liuxihe model optimization in this study.

Line 114: Replace “Terrain” by “Globally available” data.

Line 124: The authors should describe the flood peak inflow for 18 selected events selected by the authors.

Line 137-138: The authors should describe the 24-hour total catchment-scale precipitation that corresponds to 18 selected flood events in this study.

Lines 166: Figure 3 is difficult to interpret without explanation of the 3- and 4-level channels importance in the Liuxihe model flood peak calculations.

Line 188: Table 2 is difficult to interpret and needs to be explained in the view of the Liuxihe model governing equations.

Line 198: The authors must provide statistical equations that are used to judge the Particle Swarm Optimization (PSO) algorithm and reported in Table 3, 4 and 5.

Lines 199-214: This text and Figure 4 must be moved to the Results section. The authors should provide the optimization run time of PSO algorithm, see Liu  et al. (2022). In Figure 4b, what is meaning of y-axis?

Lines 217-220: This text should be moved to model setup or removed.

Line 221: This statement should be moved to Conclusions section.

Lines 231-232: The authors must rank 18 selected floods and describe the ranking of 6 representative floods shown in Figure 5.

Lines 239-240: Figure 5 caption should be revised by removing the repetitive “flood number” and changing the text as “representative floods with flood number:”

Lines 277-282: This result should be supported with a new table as demonstrated in Table 6.

Lines 284-291: This text should be moved to “Model setup” section.

Line 298: Table 5 formatting is misleading and should be removed. The authors should only keep values with the largest change between 4-level and 3-level simulations while describing other important parameters with little change in the text.

Line 303: Table 6 statistical indicators should be described by equations in the “Model Setup section” and replace the text of these indicators with values and units, for example replacing “Water Balance Coefficient” by “WBC, -“

Lines 315-316: Figure 6 caption should be revised  by removing the repetitive “flood number” and changing the text as “representative floods with flood number:”

Lines 339-343: The authors should indicate the run times of model optimization.

Lines 350-352: Robust distributed hydrological models should be able to forecast flood inflow with many upstream precipitation gauges and this statement should be revised by describing the 18 simulated floods compared with the 1000-year flood for future flood inflow forecasting with the Liuxihe model.

Reviewer 2 Report

Water 2190956 Review:

General comments:

Paper evaluates the capacity of physically distributed hydrological model (Liuxihe model) for forecasting inflow in the Fengshuba Reservoir in China. Authors demonstrate how the model parameter settings influence the simulation results. Although the paper generally shows interesting presentation of the hydrological physical model applicability for simulating river discharges during selected flood events, I found several major shortcomings that, in present form do not assure achieving a standard necessary to accept the manuscript. I found problematic especially the following points:

Overall, the results of the simulations should be analyzed more in depth in view of the model settings and specific characteristics of the studied catchment (see detailed comments).

Consideration of the spatial heterogeneity of rainfall is highlighted throughout the paper, authors state that precipitation is highly unevenly temporally and spatially distributed. However, the authors used simple spatial extrapolation of rainfall (Thiessen polygons) which is most commonly used in different types of hydrological models. In this respect, the use of spatially distributed hydrological model is not of great benefit.

Since the authors promote the use of the physically distributed Liuxihe model for forecasting purposes, it would be very informative to know what are the model computation times and what time steps were considered in the simulations. Additionally, since the physically distributed hydrological models are generally more computationally demanding compared to other hydrological models (e.g. lumped), it would be good to know how the selection of the simulation time steps influences the required model calculation times.

The authors suggest that the Liuxihe model could be used for the purpose of reservoir operation. I would strongly suggest to represent the modelling results also by considering longer simulation periods (e.g. selected multiple mothly periods when some of the mentioned floods occurred). This would in my view strongly support all the benefits of the model the authors try to highlight throughout the manuscript and are currently not addressed enough.

Specific comments:

Line 26: “The Fengshuba Reservoir was designed according to the 1,000 flood…” Do the authors here mean the reservoir storage volume or the elements of the reservoir outflow hydraulic structures (e.g. flood evacuation spillway)?

Lines 39-43: This is not entirely true. Lumped models can be (despite their relatively simplified representation of the hydrological system) in terms of structure easily adapted to specific catchment characteristics (e.g. widely used hydrological models such as Hec-HSM, HBV and others are all highly adaptable in terms of the mentioned processes). I believe in the introduction section some potential drawbacks/deficiencies of the physically based models (e.g. huge amount of input data, calibration and verification problems, problems of overparameterization) should also be mentioned.

Line 98: Why do authors consider water body as a soil type?

Line 110: How was the optimal grid cell size 90 m defined (or is it only due to the input DEM)? The land-use data and soil data have 1000m resolution, what method was used for resampling to 90m cell size?

Section 2.3: The authors highlighted the benefits of using the spatially distributed rainfall data in spatially distributed hydrological models. However, they used Thiessen polygons to consider quasi-spatially distributed rainfall data (which are widely used in various types of hydrological models including lumped ones). Why the authors didn’t explore the advantage of spatially distributed hydrological model and used spatially distributed rainfall data (if available)?

Figure 2: What do the flood number mean? It would be good to provide some information about the flood events (e.g. date when the flood events occurred, characteristics of flood hydrographs, return period of the hydrograph peak etc.).

Line 161: How were the river channel characteristics estimated?

Line 174: If I understand correctly, the spatial variations of evaporation were neglected in the model (based on the authors experiences). Since there the topography of the catchment is quite versatile, this seems as a quite strong simplification. Some additional explanations are needed to justify this. How was transpiration considered in the model?

Table 1: In what units is the evaporation coefficient and what does it mean? How many parameters are considered in the Liuxihe model? A list of all considered parameters with short description would be helpful for understanding the process of parameters’ optimization (e.g. related to the parameters presented in Fig. 4b).

Section 3.2 and 4.2: The particle swarm optimization (PSO) algorithm was used for model optimization/ calibration. How do you assure that the model is not overparameterized (large number of coefficients is mentioned) or that different combinations of parameter values lead to similar modelling results (the problem of equifinality)?

Lines 206-207: Which “certainty coefficient” is considered here? “The coefficient was 0.876” – what coefficient?

Figure 4c and figure 5: The measured flows show high temporal fluctuations, what causes these fluctuations which the model is not able to capture?

Line 242: “Uneven rainfall” in terms of spatial or temporal distribution?

Line 267: Can authors provide some additional proofs/arguments that the used parameter set is the optimal one?

Line 285-286: Why should be small watersheds divided into four levels? What sizes of watersheds are considered as small and medium size? Shouldn’t be the river classification based on river network structure (e.g. by considering mentioned Strahler method under model setup section)?

Lines 317-319: Where is this shown? Some information should be added related to the results shown in Figure 6. How the “channel level” relates to simulation results? What is the physical meaning of such channel classification?

Section 4.3 The impact of the channel classification seems huge in view of several model parameters (e.g. Manning roughness coefficient, evapotranspiration coefficient etc.). Did the authors constrain the parameter values during the parameter optimization to some physically meaningful values?

Table 5: What is the purpose of the River bottom slope (Bs) and River bottom width (Bw) parameters? Shouldn’t such parameters have a priori known/fixed values derived from the DEM data?

Figure 6: Using three-level channel or four-level channel result in substantially different shapes of the hydrographs; generally, consideration of third-level channel result in more abrupt response of the channel discharge to rainfall. Additional comments from the process point of view are needed in order to enable the readers to understand this behavior. Explaining only statistical coefficients is not sufficient in this respect.

Reviewer 3 Report

The paper is an interesting application of the Liuxihe Model in the simulation of a watershed. 

From what the authors write describing the model and from the results some questions arise, which are as follows:

-How does the model take into account the base flow of the river.

-In what way are the catchment conditions prior to the rainfall episode studied each time incorporated into the model.

The description of the model makes no mention of these questions.

The results show in all cases the value of the flow at the beginning of the simulation is equal to zero. This leads me to the conclusion that the model does not include the base flow, as well as there is no specification of any kind of initial conditions. Is this conclusion correct or am I missing something?

Round 2

Reviewer 1 Report

Thank you very much for the detailed reply to my comments and addressing these in the revised manuscript based on the track-change version. The revised manuscript has greatly improved and can accepted to the publication. However, it is hard to see the final version of the text and I suggest double-check of the final version without track-change to make sure that these changes are in the right place.  

Reviewer 2 Report

Authors have adequately answered most of the questions and comments I raised during 1st review. There are still some minor things that might be improved especially related to the following issues:

General comments:

Point 2: The real novelty of considering the spatially distributed rainfall data is highly questionable, since the authors used one of the most widely and commonly used techniques for rainfall data interpolation (Thiessen polygons). This should be more clearly noted throughout the manuscript.

Point 4: The possible limitations of the used Liuxihe model for operational purposes of the studied reservoir should be more clearly presented and highlighted.

Specific comment:

Point 12: It is a little bit weird that authors do not use as a reference data actual measured discharge data (inflows into the Fengshuba reservoir) for analysis of the model performance. I believe this should be also mentioned in the paper.

Reviewer 3 Report

The Authors have made several additions to the original text, but it is not clear whether anything has been deleted from the original text.

It is notable that the text shown on lines 22-34 is repeated on lines 63-75.

The quality of all the images has been significantly degraded.

Consequently the text should be carefully revised so that it is possible to really check whether the comments and remarks made in the initial review of the paper have been answered.

Furthermore, with regard to my comments, the authors, in the response letter submitted with their revised text, provided some explanations to the questions I raised which I do not find adequate and did not make corresponding additions to the text. 

Consequently, I expect the authors to make appropriate additions to their text that answer the questions I raised, so that they can provide the answers that readers who have the same questions as I do will need.

Round 3

Reviewer 3 Report

Authors have answered to my questions and therefore I suggest to accept the paper as it is formed.